# Paediatric Spitzoid Neoplasms: 10-Year Retrospective Study Characterizing Histological, Clinical, Dermoscopic Presentation and FISH Test Results

**DOI:** 10.3390/diagnostics13142380

**Published:** 2023-07-15

**Authors:** Astrid Herzum, Corrado Occella, Valerio Gaetano Vellone, Lodovica Gariazzo, Carlotta Pastorino, Jacopo Ferro, Angela Sementa, Katia Mazzocco, Nadia Vercellino, Gianmaria Viglizzo

**Affiliations:** 1Dermatology Unit, U.O.C. Dermatologia e Centro Angiomi, IRCCS Istituto Giannina Gaslini, Via Gerolamo Gaslini, 5-16147 Genova, Italy; astridherzum@yahoo.it (A.H.); corradooccella@gaslini.org (C.O.); lodovicagariazzocelesia@gaslini.org (L.G.); carlottapastorino@gaslini.org (C.P.); jacopoferro@gaslini.org (J.F.); nadiavercellino@gaslini.org (N.V.); gianmariaviglizzo@gaslini.org (G.V.); 2Pathology Unit, U.O.C. Anatomia Patologica, IRCCS Istituto Giannina Gaslini, Via Gerolamo Gaslini, 5-16147 Genova, Italy; angelasementa@gaslini.org (A.S.); katiamazzocco@gaslini.org (K.M.)

**Keywords:** spitz nevus, paediatric, multicomponent pattern, spitz dermoscopy, FISH

## Abstract

Introduction: Spitzoid lesions are a wide tumour class comprising Spitz nevus (SN), atypical Spitz tumour (AST) and Spitz melanoma (SM). Materials and Methods: We conducted a single-centre-based retrospective survey on all histologically diagnosed spitzoid lesions of paediatric patients (1–18 years) of the last 10 years (2012–2022). Histopathological reports and electronic records of patients were used to retrieve relevant data regarding patients’ features, clinical and dermatoscopical aspects of lesions when recorded, and FISH tests when present. Results: Of 255 lesions, 82% were histologically benign, 17% atypical, 1% malignant. Clinically, 100% of SM were large (≥6 mm) and raised; AST were mainly large (63%), raised (98%), pink (95%). Small (≤5 mm), pigmented, flat lesions correlated with benign histology (respectively 90%, 97%, 98% SN) (*p* < 0.0001). Dermatoscopical patterns were analysed in 100 patients: starburst pattern correlated with benign histology (26% SN (*p* = 0.004)), while multicomponent pattern correlated with atypical/malignant lesions (56% AST, 50% SM (*p* = 0.0052)). Eighty-five lesions were subjected to fluorescence in situ hybridization (FISH): 34 (71% AST; 29% SN) were FISH-positive; 51 (63% SN; 37% AST) were FISH-negative (*p* = 0.0038). Discussion: This study confirmed predominant benign histology (82%) of paediatric spitzoid lesions, thus detecting 17% AST and 1% SM, highlighting the need for caution in handling spitzoid lesions. Conclusion: Until AST are considered potentially malignant proliferations and no reliable criteria are identified to distinguish them, the authors suggest a prudent approach, especially in children.

## 1. Introduction

Spitzoid lesions represent a challenging entity, as their spectrum ranges from benign Spitz nevi (SN) to malignant Spitz melanoma (SM), comprising also atypical spitzoid tumours (AST), which are spitzoid tumours of uncertain malignant potential and difficult to classify histologically [1,2,3].

Spitzoid lesions were first described by pathologist Sophie Spitz in 1948 and have been widely studied morphologically and biologically ever since. At first, the term “juvenile melanoma” was used by Dr. Spitz to define these lesions, typical of childhood, with both clinical and histopathological malignant aspects. However, after observing the generally benign biological behaviour of these lesions, the spitzoid class was soon regarded as completely benign. Cases of malignancy with metastatic disease were outright attributed to misdiagnoses of melanomas mimicking spitzoid lesions [4,5,6]. In the late 1990s, spitzoid lesions with nodal metastases were described, yet without any other histological malignant criteria, and opened a new era for the characterization and classification of spitzoid lesions and of their biological behaviour [7,8,9,10].

Spitzoid lesions are all histologically characterized by spindle cell and epithelioid cell melanocytic proliferations, as first described by Dr. Spitz, but while SN display benign histological features and SM display malignant histological features, AST present some atypical histological features in between SN and SM, yet insufficient to make a diagnosis of melanoma [7].

In addition, recently, a distinction between “Spitz melanoma” and “Spitzoid melanoma” has been established. Though both present clinical and histopathological aspects reminiscent of Spitz nevi, “Spitz melanoma” presents typical melanoma driver mutations, while “Spitzoid melanoma” displays kinase fusions typifying Spitz tumours [7,11].

To ultimately distinguish spitzoid proliferations into biologically benign or malignant, sentinel lymph node biopsy (SLNB) was previously used, yet this practice is almost abandoned, as its prognostic and therapeutic utility is actually debated, in addition to its high morbidity [7,12,13].

Of note, spitzoid lesions present with an overall incidence of 1.4 to 7 cases per 100,000 person per year and characteristically develop in the paediatric age or in young adults, although all age groups can be affected. Almost 40% of Spitz lesions appear before the age of fifteen, and 77% appear before the age of thirty, while in older individuals, the presence of these lesions is quite uncommon [14,15]. By all means, among all melanocytic tumours excised during childhood, Spitz lesions represent fewer than 1% of cases [16,17].

Clinically, Spitz nevi generally appear as single dome-shaped papules, flesh-coloured to pinkish or red in colour, due to the high vascularization and low melanin content; thus, almost 10% of lesions are pigmented with brown colour [17]. The lesions characteristically grow rapidly, within 3 to 6 months, and can reach a dimension of up to 1 to 2 cm, though they generally measure 5 to 6 mm in size. After their rapid growth, lesions can remain stable for years and subsequently may show a progressive evolution towards acquiring a common melanocytic nevi appearance or towards complete involution [18]. The location of lesions is reported in the literature as typical on the head and neck (37%) in children, while adults more frequently present lower limb lesions (28%), often intensely pigmented between brown and black in adults [19,20].

According to literature data, AST are typically larger in size (>10 mm) than SN, occur in older (>10 years) patients, more often on the trunk, and are characteristically asymmetric—irregular in colour and borders. Yet further characterization and analysis of these lesions is needed [17,21].

SM are generally hypopigmented or non-pigmented lesions, though also pigmented lesions with multiple colours with progressive growth have been described in the literature. Lesions usually reach relevant (>10 mm) dimensions and may display ulceration. Head and neck and limbs are frequent locations [17,22].

Histologically, SN are well delimited melanocytic lesions, mostly with a compound presentation, that can be located both at the dermo-epidermal junction and within the dermis, though some SN are limited to the dermo-epidermal junction with cytological features similar to the compound counterpart [17]. Epithelioid cells, rounded or polygonal, or spindled melanocytes, with large but regular nuclei, presenting peripheral margination, are characteristic of SN, with prominent central nucleoli and abundant cytoplasm, containing limited melanin pigment.

The cells tend to form uniform nests, vertically oriented, with a dermal component that matures with depth and, less markedly, towards the periphery; yet a pagetoid spread of single, isolated, melanocytes can be noted at the upper epidermal layers or at the dermo-epidermal junction.

Typical mitosis (usually less than 2/mm^2^) can be observed in the upper or mid part of the lesion, while deeper mitoses are rare, and atypical mitoses are absent [14,23].

Dermal vessels are typically dilated and form clefts among the melanocytic nests. Perivascular or basal lymphocytic infiltrate can be present. Kamino bodies, characteristic of SN, can be observed in the papillary dermis and epidermis. These are eosinophilic globules containing basal membrane proteins. Kamino bodies are not specific to SN, as melanomas can also contain these globules, yet in smaller form and less structured [24]. Melanin pigment can be granulated and is mostly located in the upper part of the lesion.

AST are histologically characterized by at least one atypical feature among poor lateral delimitation, asymmetry, lack of dermal maturation, great downwards extension, plentiful isolated melanocytes in the upper dermis rather than forming nests, presence of mitoses in the dermis (usually >2/mm^2^), absence of Kamino bodies and ulceration. Although some grade of atypical features is observed in AST, criteria for diagnosing melanoma are not sufficiently met, placing AST in between SN and SM [17,21]. Notably, the presence of cytological high-grade atypia, ulceration, asymmetry and elevated number of deep mitosis have been associated with an increase in metastatic disease [25].

SM can histologically resemble SN, yet they present a high number of atypical features, higher than AST. Included among atypical features characteristic of SM are poor delimitation, asymmetry, epidermal ulceration, absence of dermal maturation with depth, extensive pagetoid spread, frequent mitoses in the dermis, absence of Kamino bodies, high grade of cellular atypia, high nucleus/cytoplasm ratio and high cellular pleomorphism [17,22].

Recently, the introduction of dermatoscopy has enabled a greater diagnostic accuracy of Spitzoid lesions through dermatoscopical observation, yet dermatoscopical criteria that are typical for SN, where they are symmetrically and regularly arranged, are not specific for only SN, but also melanomas can display these criteria, although in asymmetric, multicomponent distribution, often in combination with chrysalis like structures, though the latter can be displayed also by SN [17,22].

The most frequent pattern reported in the literature in SN is the vascular dotted pattern, characterizing up to 51% of SN and displaying monomorphous dotted vessels, regularly distributed over a pink background [10,17]. Other vascular patterns displayed by hypopigmented or non-pigmented spitzoid lesions are the vascular glomerular pattern, with coiled, tortuous capillaries; the vascular hairpin pattern, with bending, looped vessels; the vascular starburst pattern, with radial vascular lines, regularly distributed at the periphery of the lesion; and the homogeneous pink pattern, with a pink background without other relevant structures in the foreground [10,17]. In addition, a reticular depigmentation pattern, with a whitish network surrounding vessels, may be seen, and chrysalis structures, orthogonal lines of bright white colour at polarized light, are potentially present in SN.

When pigmented, SN often present a starburst pattern or a globular pattern, displaying a central homogeneous pigmentation ranging from grey to blue or brown–black in colour and with radial streaks at the periphery in the case of the starburst patten, while the globular pattern displays peripheral brown to black globules, round or oval, symmetrically and regularly distributed. In cases of diffuse homogeneous pigmentation throughout the whole lesion in the absence of other structures, the pattern is considered homogeneous. The reticular pattern displays a network of pigment similar to the one presented by other acquired melanocytic nevi. The atypical or multicomponent pattern displays an uneven distribution of colours and structures Another pattern often associated with melanoma, but possibly present also in SN, is the reticular depigmentation pattern, displaying intersecting lines around pigmented globules or around vascular dots [10,17]. Of note, if the globular pattern is not associated with a reticular depigmentation pattern, it is not considered characteristic of SN [26]. Indeed, oval or round structures at the periphery of melanocytic lesions are common in children’s acquired melanocytic nevi [27]. In SN, a sequential course of dermatoscopical patterns has been observed and described in the literature; starting from a globular pattern, the SN might go through different evolutionary stages, displaying a starburst pattern and a homogeneous and reticular pattern before involuting and spontaneously disappearing in the majority of cases [18].

The multicomponent pattern was reported to be highly associated with the majority of AST, and also the dotted vascular pattern was associated with AST; 16% of AST were observed to display a vascular dotted pattern [28].

SM must be considered in cases of an asymmetry of typical spitzoid features, in cases of a multicomponent pattern, and in the presence of chrysalis-like structures [10,17].

Overall, due to the wideness and the complexity of the spitzoid class, management difficulties arise in daily practice, calling for the necessity of shared handling approaches; yet the literature provides conflicting guidance on the best practice to adopt for the management of spitzoid lesions [4,6]. Especially in the paediatric field, the biological behaviour of spitzoid lesions is highly debated, and many succeeding guidelines have been recently established for their management [7,8,9,10,29]. A conservative approach is often suggested in the management of spitzoid lesions in children, especially if patients are younger than twelve years old, considering the rarity of dermatological malignant lesions in children and the even rarer mortality caused by SM in young children, that, indeed has a more favourable prognosis than the adult counterpart of Spitz lesions [1,7]. Yet, paediatric dermatologists are encouraged to err on the side of caution according to recent literature data reporting histological features of SM in up to 0.5% of paediatric spitzoid lesions and histological features of AST in up to 17% [7]. Indeed, in paediatric patients older than twelve, the current international dermoscopy society (IDS) guidelines suggest excision or close monitoring of all spitzoid lesions also in the absence of atypical features [10,29], and even expert dermoscopists suggest erring on the side of caution when it comes to Spitzoid lesions in older children (>12 years), as even clinically and dermoscopically banal looking lesions may have aggressive biological behaviours and aggressive disease courses, not to mention the lesions that present with atypical features, clinical or dermoscopical, where excision is mandatory at every age [10].

Considering the high incidence of spitzoid lesions in children, paediatric dermatologists and pathologists first, but in general all dermatologists, pathologists and general practitioners, are called to improve their understanding and knowledge about this heterogeneous class of tumours.

Herein we present a large series of paediatric-specific spitzoid neoplasms from a single centre with the aim of analysing the epidemiological, morphological and genetic aspects of paediatric spitzoid lesions.

## 2. Materials and Methods

At the Dermatology Unit of the Paediatric Hospital IRCSS Istituto Giannina Gaslini, we conducted a retrospective survey on all histologically diagnosed spitzoid lesions of paediatric patients (1–18 years) of the last 10 years (2012–2022). Histopathological reports and electronic records of patients were used to retrieve all relevant data: histopathological diagnosis, age at diagnosis, sex, anatomical site of lesion, size of lesion, clinical aspect, dermatoscopical aspect if recorded and FISH test when present. All lesions were surgically excised with punch biopsy or with fusiform excision using a no. 15 scalpel blade. Then standard histological examination was carried out by two expert dermatopathologists trained in skin tumour diagnosis on formalin-fixed, paraffin-embedded tissue sections. Histologic sections were obtained dissecting the specimen perpendicular along its longest axis. Conventional histology using a routine staining procedure with haematoxylin and eosin (H/E) staining was applied, and immunohistochemical markers, including MIB-1, HMB-45, p16, S100 and Melan-A, were used as complementary analyses to the ultimate histopathological diagnosis.

Lesions displaying overlapping histologic features with conventional Spitz nevus and spitzoid melanoma were defined as AST, a borderline spitzoid melanocytic lesion of uncertain malignant potential.

Fluorescence in situ hybridization (FISH) conventional melanoma probes (Vysis/Abbott Molecular, Des Plaines, IL, USA) with probes targeting four loci—cyclin D1 (CCND1) on 11q13 (Vysis LSI CCND1-Spectrum green), V-myb myeloblastosis viral oncogene homolog (MYB) on 6q23 (Vysis LSI MYB-Spectrum gold), ras responsive element binding protein 1 (RREB1) on 6p25 (Vysis LSI RREB1-Spectrum red) and centromeric enumeration probe control for chromosome 6 (Vysis LSI CEP6-Spectrum aqua) were performed in all histologically diagnosed AST and in an equal number of SN, matched for age and sex, as controls. The FISH test was considered positive for the following chromosomal abnormalities: greater than 38% CCND1 gain; greater than 29% RREB1 gain; greater than 55% relative gain of RREB/CEP6; and greater than 40% relative loss of MYB/CEP6.

Patients with histologically atypical and malignant lesions were followed up for clinical evaluation of local recurrences and lymph node macro-metastases.

### Statistical Analysis

Fisher’s exact test and χ^2^ test were used to analyse data statistically; *p* values < 0.01 were considered statistically significant.

## 3. Results

Overall, 255 paediatric spitzoid lesions were included in the present study, with mean age of 7.9 years, median age of 7 years, and male:female ratio of 1.1:1. Histologically, 82% were benign (210 SN), 17% atypical (43 AST), and 1% malignant (2 SM) (Figure 1 and Figure 2). Features of patients and clinical features of spitzoid lesions are reported in Table 1. Neither gender nor increasing age range were statistically associated with malignancy of lesions (*p* > 0.05). Specifically, patients 12 years or older had no more malignant lesions (50 SN; 7 AST/SM) than younger patients (160 SN; 38 AST/SM) (*p* > 0.05).

Concerning the anatomic distribution of spitzoid lesions, lower limbs were most frequently involved (38%: 30 thigh, 30 leg, 18 knee, 13 gluteal, 6 ankle), followed by upper limbs (21%: 26 arm, 16 forearm, 4 shoulder, 4 wrist, 3 elbow), head and neck (21%: 38 face, 8 ear, 6 neck, 1 scalp), trunk (12%: 24 dorsum, 4 abdomen, 2 chest), hands and feet (7%: 10 hand, 9 foot) and perineum (1%: 1 scrotal, 1 inguinal, 1 intergluteal). Of SN, 35 were identified on the head and neck, 27 on the trunk and 148 on the limbs (Table 2). Notably, while lower limb lesions were statistically associated with benign histology (89% SN) (*p* = 0.0385), head and neck were more associated with atypical/malignant histology (34% AST/SM) (*p* = 0.0005).

Regarding the clinical aspect of lesions, 54% were hypo-/non-pigmented vs. 46% pigmented; 76% were raised vs. 24% flat. Overall, 100% of SM were large (≥6 mm) and raised (100%); 50% were pink; AST were mainly large (63%), raised (98%) and pink (95%). Conversely, small (≤5 mm), pigmented and flat lesions were statistically associated with benign histology (90%, 97% and 98% SN, respectively) (*p* < 0.0001) (Table 1). Head and neck SN were statistically associated with hypo-/non-pigmentation (74%) (*p* = 0.0003; OR 4.33); while SN of the limbs were associated with pigmentation (62%) (*p* = 0.0005; OR 2.99) (Table 2). Interestingly, the simultaneous presence of pigmentation, flatness and lower limb location was evidenced in 19 cases (Figure 3). All of them (100%) were benign.

Interestingly, flat lesions were statistically more associated with the presence of pigmentation (92% (55/60) pigmented; 8% (5/60) hypo-/non-pigmented) than raised lesions (32% (62/195) pigmented; 68% (133/195) hypo-/non-pigmented (*p* < 0.0001); OR 66.50, 95% CI 25.97–170.27).

Dermatoscopical patterns were retrieved and analysed in 100 patients (62 SN, 36 AST, 2 SM). The retrieved patterns are displayed in Table 3; vascular (20%), starburst (16%), globular (8%), multicomponent (37%), reticular (4%), aspecific (8%) and homogeneous (7%) patterns were displayed by spitzoid lesions (Table 3). Of note, a starburst pattern, corresponding to Reed nevus, was retrieved in 16 SN (26%) and in 0 AST/SM (*p* = 0.004). Conversely, a multicomponent pattern proved an association with atypical/malignant lesions (20 AST (56%) and 1 SM (50%), respectively; *p* = 0.0052). No other dermatoscopical pattern proved a statistical association with histology (*p* > 0.05) (Table 3); 17% of AST as well as 23% of SN had similar vascular patterns, almost indistinguishable (Figure 4 and Figure 5); 17% of AST and 3% of SN had aspecific patterns.

Finally, FISH analysis was performed in 85 lesions, comprising all 43 histological AST cases and 42 histological SN controls, matched for age and sex. Overall, 34 proved positive with the FISH test (71% AST; 29% SN), and 51 were FISH test negative (63% SN; 37% AST). FISH test-positive lesions correlated with atypical histology (*p* = 0.0038; OR 4.04) (Table 1).

Regarding clinical follow-up and further management of patients, patients with AST were clinically evaluated every six months. SM were treated according to melanoma protocol. Among the numerous AST lesions analysed in this study, only 1 (2%) AST presented clinically evident lymph node macro-metastasis, and no clinical recurrences were reported.

## 4. Discussion

This single centre study comprised over 250 consecutive paediatric-specific spitzoid neoplasms of the last ten years, permitting us to obtain a wide case series of paediatric patients. Overall, literature data were confirmed. Paediatric spitzoid lesions were histologically benign in 82% of cases, though a relevant percentage of AST (17%) were retrieved, as well as 1% of SM, highlighting the need for utmost caution in the management of spitzoid neoplasms [1,2,3,4,5,6,7]. Recently, Bartenstein et al. reported comparable data in their retrospective cohort study on young (<20 years) patients with a histopathologic diagnosis of spitzoid proliferation: 82.3% were typical, 17.2% were atypical, and 0.5% were melanomas [7]. In addition, Davies et al. described similar data in their retrospective single centre study on spitzoid lesions, though with higher prevalence of benign lesions (91%) and lower prevalence of AST and SM than in our cohort (8% and 0.3%, respectively) [8].

At present, the management of spitzoid lesions is controversial, and numerous succeeding guidelines have been developed by the IDS to help clinician handle morphological overlap between SN, AST and SM [10].

In young children (<12 years) a conservative approach is suggested in the management of spitzoid lesions, considering the rarity of malignant spitzoid lesions in young children and their utmost favourable prognosis [1,7,10].

Conversely, in a study by Lallas et al. conducted on 384 patients aged twelve years or older presenting spitzoid lesions, the risk of melanoma was around 13%. Notably, the risk increased with increasing age, reaching 50% risk of melanoma in patients aged 70 [30]. Consequently, excision or close monitoring of all spitzoid lesions in patients 12 years or older is suggested, as well as excision of clinically or dermoscopically atypical spitzoid lesions at any age [10].

However, it must be considered that IDS guidelines were developed relying on studies including mainly adult patients, with a mean age of 27.7 years. Thus, adult patients may represent a statistical bias, with underreported paediatric patients [10]. On the other hand, the actual study focuses expressly on paediatric patients (mean age 7.9 years), allowing for greater statistical power in this age group. This is of relevance, considering that spitzoid lesions characteristically develop in the paediatric age [3]. Moreover, 78% of presently studied patients were <12 years, and of these, almost one fifth (19%) carried an atypical or malignant lesion.

Nevertheless, IDS guidelines further classify spitzoid lesions among young children; in patients <12 years, a symmetric starburst pattern can be handled conservatively, as it corresponds to Reed nevi [10]. IDS 2018-updated guidelines do not even warrant any further action for such lesions, as this would lead only to unnecessary excisions without any actual benefit [29]. Data from the present study support these criteria, also including paediatric patients >12 years; clinically pigmented, flat and lower limb lesions were respectively benign in 97%, 98% and 89% of cases. The simultaneous presence of these criteria was always associated with benign histology (100%), thus possibly representing a criterion for conservative management in paediatric patients. In addition, the presence of a dermatoscopic starburst pattern could represent the final clue to assessing lesions corresponding to Reed nevus and benign histology (*p* = 0.004; OR 27.32).

Contrarily, a dermatoscopical multicomponent pattern and nodular spitzoid lesions should raise high suspicion in spitzoid paediatric lesions, being statistically associated with histologically atypical/malignant lesions (respectively, 57% and 23%) and demand utmost caution, as already suggested by Moscarella et al. [28].

Nonetheless, it is worth noting that the malignant potential of AST has been widely discussed lately, as patients, especially paediatric patients, tend to have a favourable prognosis, even after positive SLNB [1,4], while the positivity rates of SLNB are higher in younger patients, though with better prognosis, than in older patients [31].

Notably, in a cohort of 67 patients with AST, Ludgate et al. observed 47% positive for sentinel lymph nodes, yet of 57 patients subjected to SLNB, no patients presented recurrence of disease, and at 43.8 month follow-up, all of them were disease free [4]. Indeed, the present study also confirms this; among the numerous (43) AST lesions analysed, no local recurrences were reported, highlighting the necessity of gaining more insight into the biologic role of AST [32,33,34].

However, the current definition of AST, as tumours of uncertain malignant potential, necessarily demands the highest attentiveness. Neither difficult-to-treat anatomical sites nor concerns about anaesthesiologic procedures should by any means constitute a therapeutic limitation in children [35]. In paediatric dermatology units, surgical excisions in children are routinely performed, also in special sites. Of relevance, the present study highlights the need for extra caution, especially in handling lesions localized on the head and neck, which are frequently (34%) histologically atypical/malignant and where hesitations about aesthetic outcomes should not represent a diagnostic limit.

Regarding the clinical application of cytogenetics, melanoma probes FISH technology recently demonstrated a high impact on melanocytic skin lesion analysis, permitting the rapid screening between melanomas and nevi [36,37] but not yet in the actual grey zone represented by AST.

Indeed, Gerami et al. proposed melanoma probe FISH to detect the copy number of genes RREB1, MYB, CCND1 and CEP6 [37]. While the presence of abnormalities would point towards melanoma, the absence of abnormalities points towards Spitz nevi. In between, AST lesions with aggressive biological behaviour also exhibit the presence of abnormalities [37].

However, the majority of studies on the clinical applicability of FISH did not present long-term clinical follow up [37,38,39,40]. So far, it has only been hypothesized that AST may show fewer molecular changes than melanoma, and more than SN, but previous studies using the melanoma FISH probes did not show relevant results [13,26].

In the present study, FISH-positive lesions correlated with atypical histology (*p* = 0.0038; OR 4.04) (Table 1), possibly suggesting a potential role of melanoma FISH probe analysis in screening AST among paediatric histologically spitzoid lesions.

Recently, FISH probes with greater specificity for spitzoid lesions than melanoma FISH probes were also developed [41]. These FISH probes target genes RREB1 (6p25), CCND1 (11q13), MYC (8q24) and CDKN2A (9p21) [41]. The loss of 9p21 in homozygosis was associated with aggressive biological behaviour [42].

Molecular investigations represent promising complementary diagnostic tools to characterize spitzoid lesions and to predict their malignant potential, yet high costs and low availability have so far hindered real-world use in daily clinical practice. Universal, wider population studies would help to gain a deeper understanding and to standardize these techniques.

## 5. Conclusions

Spitzoid lesions are highly difficult-to-manage lesions, especially in children, considering the complexity of their clinical, dermoscopical and histological differentiation [1,2,3,4] In addition, their prognosis and malignant biological potential are very diverse and considered uncertain for paediatric AST [1,2,3,4]. Especially large, raised, pink lesions and a dermatoscopical multicomponent pattern should raise high suspicion in spitzoid paediatric lesions, as they were statistically associated with AST. On the contrary, pigmented, flat, lower limb lesions might be conservatively managed in paediatric patients, as all of these were consistently benign.

More and more evidence is provided in the literature of the low biological aggressiveness of paediatric AST [19]. Yet the authors suggest a prudent approach, especially in children where prevention standards must be extremely high, at least until AST are considered potentially malignant, and no clinically reliable criteria are identified to distinguish among spitzoid proliferations.

## Figures and Tables

**Figure 1 diagnostics-13-02380-f001:**
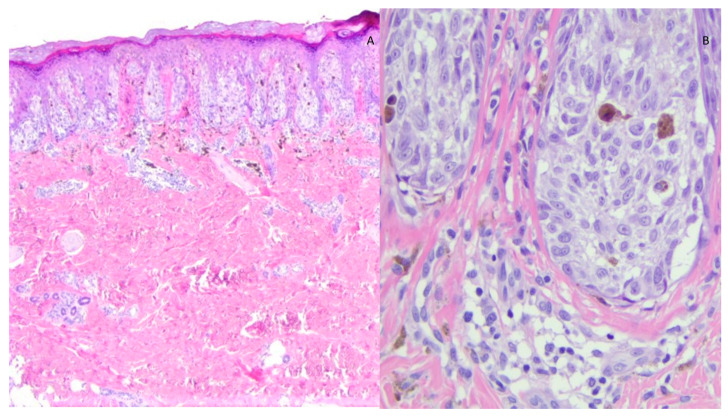
Histological image of SN showing (**A**) symmetrical, circumscribed junctional melanocytic proliferation consisting of often vertically oriented nests (raining down appearance) (EE 2×); (**B**) the junctional nests are composed of elements with spitzoid morphology: large, weakly basophilic cytoplasm and large nucleus with prominent nucleolus (EE 40×).

**Figure 2 diagnostics-13-02380-f002:**
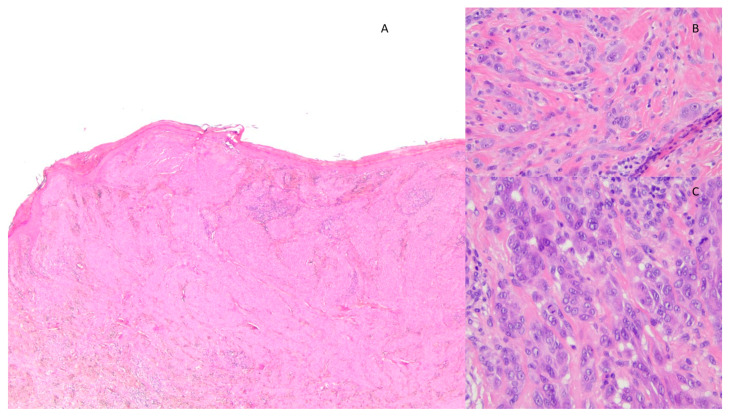
Histological image of AST showing (**A**) compound melanocytic proliferation characterized by architectural asymmetry and hypercellular, irregular intradermal growth reaching the dermo-hypodermal border (EE 2×); (**B**) elements of spitzoid morphology with atypia such as anisonucleosis and nuclear pleomorphism (EE 40×); and (**C**) several mitoses in the deep dermal component (EE 40×).

**Figure 3 diagnostics-13-02380-f003:**
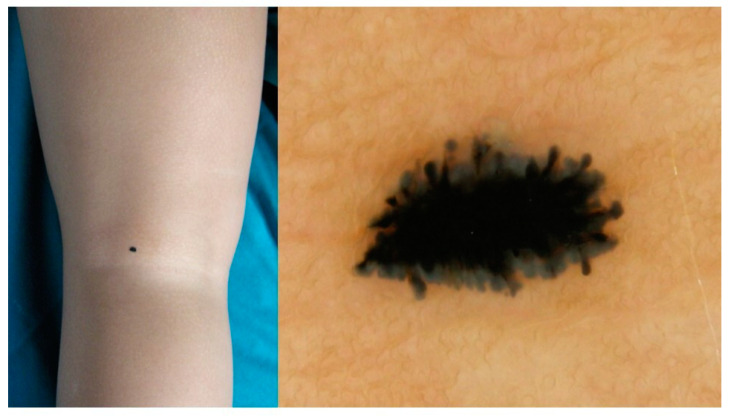
Clinical and dermoscopical aspects of Reed nevus of the lower limb: intensely pigmented flat lesion exhibiting starburst pattern.

**Figure 4 diagnostics-13-02380-f004:**
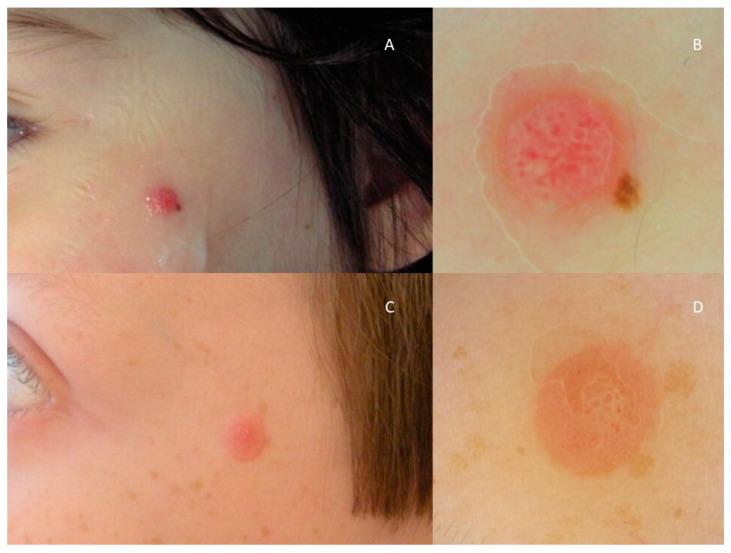
Clinical and dermoscopical aspects of histologically confirmed AST (**A**,**B**) and SN (**C**,**D**) of the face: both presenting as pink, papular lesions, dermoscopically exhibiting vascular patterns of clods.

**Figure 5 diagnostics-13-02380-f005:**
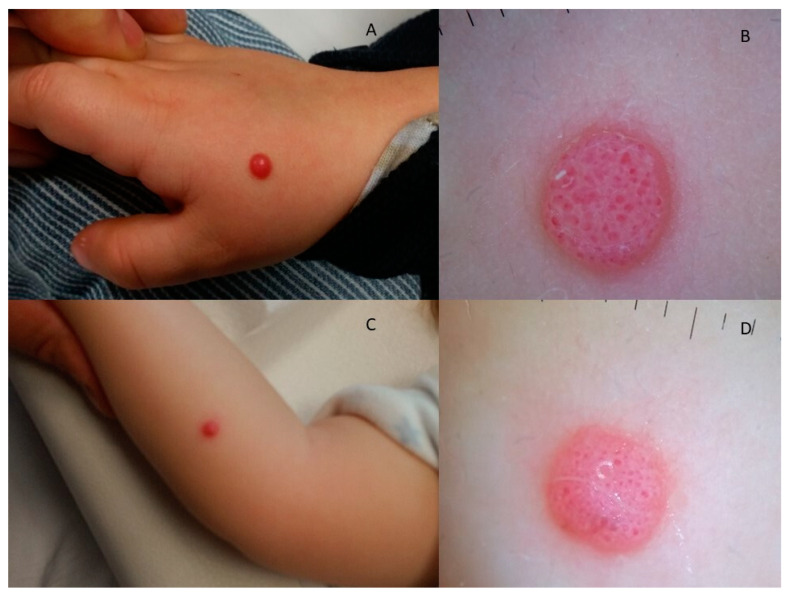
Clinical and dermoscopical aspects of histologically confirmed AST (**A**,**B**) and SN (**C**,**D**) of the upper limbs: both presenting as pink, papular lesions, with vascular patterns of clods.

**Table 1 diagnostics-13-02380-t001:** Features of patients and clinical characteristics of studied spitzoid lesions.

	Total of Lesions (%)	Benign (%)	Atypical (%)	Melanoma (%)	*p*-Value	OR (95% CI)
NUMBER OF LESIONS	255 (100)	210 (82)	43 (17)	2 (1)		
SEX						
female	123 (48)	97	25	1		
male	132 (52)	113	18	1		
mean (median) age at diagnosis in years	7.9 (7)	8 (7)	7.1 (7)	11.5 (11.5)	>0.05	
age ≤ 11 years	198	160 (76)	37 (86)	1 (50)		
age ≥ 12 years	57	50 (24)	6 (14)	1 (50)		
mean (median) lesion dimension in mm)	5.3 (5)	5 (5)	6.3 (6)	7.5 (7.5)		
Diameter 1–5	158 (62)	142 (68)	16 (37)	0 (0)	<0.0001	3.7849 (1.9264–7.4364)
Diameter 6–12	97 (38)	68 (32)	27 (63)	2 (100)		
LOCALIZATION	255 (100)	210 (82)	43 (17)	2 (1)		
Hands and feet	19 (7)	17 (8)	2 (5)	0 (0)	>0.05	
Head and neck	53 (21)	35 (17)	17 (40)	1 (50)	0.0005	3.3333 (1.6583–6.7001)
Trunk	30 (12)	27 (13)	2 (5)	1 (50)	>0.05	
Upper Limb	53 (21)	42 (20)	11 (25)	0 (0)	>0.05	
Lower Limb	97 (38)	86 (41)	11 (25)	0 (0)	0.0385	2.1437 (1.0296–4.4635)
Perineum	3 (1)	3 (1)	0 (0)	0 (0)	>0.05	
SHAPE	255 (100)	210 (82)	43 (17)	2 (1)	<0.0001	17.1921 (2.3154–127.6519)
Raised	195 (76)	151 (72)	42 (98)	2 (100)		
Flat	60 (24)	59 (28)	1 (2)	0 (0)		
COLOR	255 (100)	210 (82)	43 (17)	2 (1)	<0.0001	16.6250 (4.9955–55.3284)
Pigmented	117 (46)	114 (54)	2 (5)	1 (50)		
Hypo-/non-pigmented (=pink)	138 (54)	96 (46)	41 (95)	1 (50)		
FISH	85 (33)	42 (20)	43 (100)	0 (0)	0.0038	4.0421 (1.5935–10.2534)
Positive	34	10 (29)	24 (71)	0 (0)		
Negative	51	32 (63)	19 (37)	0 (0)		

**Table 2 diagnostics-13-02380-t002:** Characteristics of SN reported in the study.

	Total of Benign SN Lesions	Hypo-/Non-Pigmented (=Pink Lesion, Pigmentation Involving <50% of the Lesion)	Pigmented (Pigmentation Involving ≥50% of the Lesion)	*p*-Value	OR (95% CI)
NUMBER OF LESIONS	210	96	114		
Head and neck	35 (17)	26	9	0.0003	4.33 (1.9159–9.8008)
Trunk	27 (13)	14	13	>0.05	
Limbs	148 (70)	56	92	0.0005	2.9870 (1.6112–5.5375)

**Table 3 diagnostics-13-02380-t003:** Global dermoscopic patterns of dermoscopically evaluated lesions.

	Total of Lesions (%)	Benign (%)	Atypical (%)	Melanoma (%)	*p*-Value	OR (95% CI)
GLOBAL DERMATOSCOPIC PATTERNS	100 (100)	62 (62)	36 (36)	2 (2)		
VASCULAR Dotted or clods vessels on pink background	20 (20)	14 (23)	6 (17)	0 (0)	*p* > 0.05	
STARBURST (REED) Symmetric centrifugal streaks/large globules	16 (16)	16 (26)	0 (0)	0 (0)	*p* = 0.004	27.3226 (1.5872–470.3542)
GLOBULAR Brown/black globules mainly	8 (8)	6 (10)	2 (6)	0 (0)	*p* > 0.05	
MULTICOMPONENT Combination of ≥2 different predominant features	37 (37)	16 (26)	20 (56)	1 (50)	*p* = 0.0052	3.5515 (1.5091–8.3582)
RETICULAR Pigment network mainly	4 (4)	4 (6)	0 (0)	0 (0)	*p* > 0.05	
ASPECIFIC Main pattern not referable to any category	8 (8)	2 (3)	6 (17)	0 (0)	*p* > 0.05	
HOMOGENEUOS Homogeneous pigmentation mainly	7(7)	4 (6)	2 (6)	1 (50)	*p* > 0.05	

## Data Availability

The data that support the findings of this study are available from the corresponding author (A.H.) upon reasonable request.

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
