# Peer review of "Paediatric Spitzoid Neoplasms: 10-Year Retrospective Study Characterizing Histological, Clinical, Dermoscopic Presentation and FISH Test Results"

_diagnostics, 2023, doi:10.3390/diagnostics13142380_

Round 1

Reviewer 1 Report

I thank the academic editor for giving me the opportunity to review this manuscript. Rarely, in the course of my activity as a reviewer, have I given high evaluations, but in this case I feel I have to make an exception, as the paper is a really good example of a retrospective study in which the authors carry out an analysis of a cohort of 255 lesions from patients ranging in age from 1-18 years, with lesions ranging from Spitz Nevus (SN), Atypical Spitz Tumor (AST) to Spitzoid Melanoma (MS) and Spitz Melanoma. There is no important note that I have to make, everything is very clear and understandable even to the less expert reader, but, below, I specify only a few suggestions to make the paper even more attractive. 1. line 48: Not epitheliod but epithelioid. 2. line 103: not supper but upper. 3. line 190: #15 scalpel blade must be written in a "normal" style. 4. I suggest to add a small paragraph titled "Statistical analysis" with the proper informations. 5. Line 226: not nodes but nests. 6. Figure 1 (A): better contrast please. Figure 2 (A): low magnification please.

Author Response

Authors’ reply:

We thank the reviewer for the positive remarks and for the suggestions.

We corrected:

  1. line 48. epithelioid.
  2. line 103: upper.
  3. line 190: #15 was substituted with nr. 15
  4. A paragraph titled "Statistical analysis" was added, with the proper information.
  5. Line 226: nests.
  6. Figure 1 (A): the contrast was enhanced
  7. Figure 2 (A): the figure was zoomed out

================================================

Reviewer 2 Report

I find the article successful both in terms of the writing and the subject matter. 

Author Response

We thank the Reviewer for the positive remarks and are glad he found the manuscript interesting.

================================================

Reviewer 3 Report

I thank the academic editor for giving me the opportunity to review this very interesting manuscript entitled "Pediatric spitzoid neoplasms: 10-year retrospective study characterizing histological, clinical, dermoscopic presentation and FISH-test results" in which the authors report their 10-year experience relating to such an interesting and, in some ways controversial, topic relating to spitzoid neoplasms in children. I have no particular comments as I believe that the study is well conducted , with a very broad and clear introductory part, a detailed section of materials and methods, results and with a complete discussion.I recommend only to check for some typos and English language.

minor

Author Response

We thank the Reviewer for the positive remarks anf for highlighting the need for checking on typos and English language.

We corrected :

line 48. epithelioid.

line 103: upper.

line 226: nests.